# Measles at Work: Status of Measles Vaccination at a Multinational Company

**DOI:** 10.3390/vaccines7010008

**Published:** 2019-01-16

**Authors:** Nora Moussli, Emmanuel Kabengele, Emilien Jeannot

**Affiliations:** 1Institute of Global Health, Faculty of Medicine, University of Geneva, 1202 Geneva, Switzerland; nora.moussli@gmail.com (N.M.); Emmanuel.Kabengele@unige.ch (K.E.); 2Service of Community Psychiatry, Lausanne University Hospital, 1011 Lausanne, Switzerland

**Keywords:** Measles, immunization status, vaccination

## Abstract

Background: This study aims to evaluate the status of measles vaccination among employees working for a multinational company. It also assesses the effectiveness of an on-site prevention campaign. In keeping with the guidelines of the World Health Organization regarding measles awareness, the Federal Office of Public Health in Switzerland aims to eliminate measles by 2020. Methods: A questionnaire about measles vaccination was sent by e-mail and via a fluid survey. Logistic regression models examined the associations between explicative variables and the status of complete measles immunization. The status of complete measles immunization was used as the primary outcome. Results: 17% of the participants were not aware of their measles immunization status, 14% had had only one dose of the vaccination, and only 24% had two doses. Male employees had a lower probability of being vaccinated against measles than women [aOR = 0.62; 95% CI: 0.43–0.86]. Employees of Swiss and African origin had a higher probability of being vaccinated than employees of European origin (aOR = 1.94; 95% CI: 1.13–3.33). Conclusions: Based on the results of the questionnaire, further efforts are needed to promote measles vaccination through awareness campaigns so that employees become more aware of the importance of measles immunization.

## 1. Introduction

Measles was a highly contagious disease caused by the measles virus. The virus is transmitted in droplets when people cough or sneeze. Measles was not specifically a childhood disease; it is no longer a childhood disease in countries with high measles containing vaccine coverage. It can occur at any age [1]. Adults older than 20 years of age are more likely to suffer from complications [2]. Uncomplicated cases may heal fairly rapidly without leaving any sequela or side effects. However, the following and other complications can occur: Encephalitis (1 in 1000 cases), measles-related pneumonia (10 to 60 in 1000 cases), or middle ear infections [3].

### 1.1. Measles Swiss Context

The measles vaccination coverage has increased in children of all age groups since 2000 in Switzerland, and nevertheless, the national target of 95% of people vaccinated with two doses has not yet been reached, this national target follow Global Vaccine Action Plan (GVAP) of the WHO. During the period 2014–2016 the national coverage for a single dose was 94–96% depending on the age group for children (six–eight years), but this coverage is only 87–93% for two doses. Only a few cantons achieve the goal of 95% coverage for two doses. Conveniently, vaccination coverage of adults is not the objective routinely monitored in Switzerland and we do not actually have solid fact about it [4]. Between 2006 and 2009, the country experienced a large outbreak of measles with 4400 cases among children and adults [5]. During January and August 2011, there was one other large measles outbreak with 219 cases (47 cases per 100,000 inhabitants happened in the canton of Geneva, the majority of cases were not (81%) or incompletely vaccinated (8%). About one fifth of cases (44 cases) were imported or related to imported cases [6]. Switzerland has also been responsible for exporting cases of measles, particularly in Columbia at the end of 2015 and to Australia in early 2016 [7]. In 2018, 48 cases of measles were reported in Switzerland—principally adolescents and young adults [8].

### 1.2. Multinational Company Studied Context

The company chosen for this study was a multinational company and one of the largest employers in the canton of Geneva. The employees enrolled travel regularly and extensively all over the world, and therefore, at a greater risk of catching or transmitting measles. Updating measles vaccinations for the employees of this company that often travels was not only important for the elimination of measles in employees home countries, but also to prevent the spread of measles to regions where measles have previously been eliminated, or in regions with low endemicity [9]. 

The aim of this study was to evaluate the vaccination status among employees working in a multinational company in Geneva and to promote a measles prevention campaign within the company.

## 2. Methods

This study was conducted in June 2017. A questionnaire was used, based on the "Fluid Survey tool", which was an on line survey tool (https://fluidsurveys.com/). The questionnaire was made up of six closed questions, which could be answered by the employees in about five minutes. It was sent confidentially to each of the employees The questions focused on gender, country of birth, contraction of the disease, date of birth [before or after 1964―as no vaccine was available before this date], possession of a vaccination records booklet and information concerning the number of measles vaccine doses received. For this data, employees indicated the number of doses of measles vaccine included in their immunization booklet. If they had any doubt, they could check their vaccination booklet with the company nurse. 

The survey took four months, after a prevention campaign, which was implemented as a follow-up to the survey. This prevention campaign included posters and displaying plasma screens throughout the company’s headquarters. Prevention against measles was also done by the company’s in-house doctor during his consultations with employees preparing their professional travel overseas. 

### 2.1. Statistical Analysis

To examine the variable distribution, frequencies and percentages for categorical variables were computed and an analysis was conducted with the exact Pearson chi-square tests for categorical variables. Logistic regression models examined the associations between explicative variables and the status of complete measles immunization. The status of complete measles immunization was used as the primary outcome. In multivariable models, only those covariates were included, which were of a priori interest of univariate analysis. All the tests were two-sided and results were considered significant at 0.05. Statistical analyses were performed using STATA 11.0 (Stata Corp, College Station, TX, USA) for Windows.

### 2.2. Ethical Approval

This study was approved by the company’s Human Resources Department and was performed in accordance with the company’s confidentiality policies. In addition, the study was accepted by the Cantonal Health Service, General Directorate for Health in Geneva, Switzerland. (Reference number CEER 2016-0238). This study was conducted in accordance with the Swiss law, as well as in accordance with the recommendations of Good Clinical Practices (ICH E6-1996) and the Declaration of Helsinki (Fortaleza, October 2013).

## 3. Results

During this study, 575 workers completed the web questionnaire (participation rate 55%). The participants’ baseline characteristics were presented in Table 1. The average age of respondents to the company wide survey was 37 years of age. About the respondents, 55% were women and 45% men. Regarding the origins of participants, 11% were born in Switzerland, 74% in Europe, 5% in Africa, 4% in Asia, 4% in North America, and 3% in South America. Thirty-six percent of participants declared that they had contracted the disease, 44% said they haven’t contracted measles disease, and 20% did not know if they had or not.

Vaccination coverage rate for the sample was 24.3% [95% CI: 20.9–27.9] for two doses of vaccines and 13.9% [95% CI: 11.2–16.9] for only one dose of vaccine. An additional 17% stated that they had never received any vaccinations (*p* < 0.001). These results demonstrate that 45% of participants were not aware of their vaccination status.

Different rates of immunization coverage were observed by gender and nationality. Forty-three percent of the women in the sample had received at least one dose of vaccine against only 31% of men (*p* = 0.019). Fifty percent of Swiss employees reported having received at least one dose of vaccine compared to 34% for those of European origin. Coverage rates on other continents fluctuated between 45% and 50%. 

Multivariate analysis presented in Table 2 indicate that male employees had a lower probability of being vaccinated against measles than women [adjusted OR = 0.62; 95% CI: 0.43–0.86]. Employees of Swiss and African origin had a higher probability of being vaccinated than employees of European origin (adjusted OR = 1.94; 95% CI: 1.13–3.33) and (adjusted OR = 2.26; 95% CI: 1.01–5.05). Another predictive factor analyzed showed that employees born after 1964 were much more likely to be vaccinated than those born before 1964 (adjusted OR = 3.28; 95% CI: 1.42–7.58).

## 4. Discussion

Following the outcome of the survey, further information was provided to the employees who participated in the study, as well as to others working for the company to ensure the maximum protection against measles. A follow-up communications campaign was also realized through the use of on-site plasma screens, toilet teasers, and posters. Employees were also redirected to their respective private doctors, gynaecologists for women, Geneva University Hospital and primary care centres. Many employees contacted the medical staff on site to clarify their personal situation. Employees who did not know their vaccination status were recommended to have a blood test done by their private doctor in order to check their immunity against measles and to be vaccinated if necessary. The information campaign stressed that unvaccinated persons pose a danger to those who cannot be vaccinated because of their age or for medical reasons. Getting vaccinated against measles was, therefore, an act of solidarity that protects people who are particularly at risk. Following the communications campaign, 23 employees updated their immunization status via the company doctor. 

The last epidemic of measles resulted in economic costs of about CHF 6.8 million in direct medical expenses in Switzerland. If indirect costs were added (i.e., the loss of productivity in both the economy and in households), the total economic impact of this epidemic totalled more than CHF 15 million (excluding public health measures). To underscore the importance of the measles vaccination, the losses referred to above were substantial when compared to the cost to Switzerland of a measles patient (between 3600 francs and 5000 francs on average; that of a vaccination with two doses of MMR vaccine, which is about 140 francs) [10]. 

Based on these figures, it can be seen that the optimal protection of employees could avoid unnecessary additional costs not only to the company in term of absenteeism, health care, and productivity, but to the national economy where that company is based, in this case, Switzerland.

Considering the large number of employees in the company studied, and the fact that so many of them regularly travel all over the world, the subject of vaccination was particularly important and needs to be made better known and be reinforced with an adequate prevention program. This was even more important considering the human and economic costs at both company and national level, which could easily be saved. 

### Limitations of the Study

While the study led to specific results, there were some unavoidable limitations in its execution. The study was carried out in only one company, and secondly, it was based on self-reporting by employees who, at the time they took the survey, most likely did not have access to their medical records or vaccination booklets ready at the time of employees reply to the survey. In absence of a blood test, we cannot be sure of their measles immunological status. We do not know for those who reported being infected with measles or the date of their infections (adult or child, during travel, or other possibilities). Another limitation is that for these employees, we do not know if their reported infection is effectively measles, or another disease with the same clinical signs as dengue, for example. The low number of participants for some category (for example, employed from African origin) means that the interpretation of statistical analysis must be done with great caution.

This lack of input could lead to biases. This study was carried out in a homogeneous population with a higher socio-economic status, meaning there was a lack of diversity in the sample. 

## 5. Conclusions

In response to these findings, considerable efforts are still needed to ensure better protection against measles epidemics within the company. The saving costs of measles prevention both for the company and at a national level must be taken into consideration [10]. One way would be to consider a control of blood immunity when all new hires start to work in the company. Catch-up vaccinations, with up two doses of the MMR vaccine, remains relevant for all subjects who do not know their measles vaccination status, especially adults born after 1963 [11].

It seems evident that a secure way to assert immunity against measles was to observe the doses of two vaccines administered in the vaccination record or to carry out an individualized serology. This study on measles could be applied to other diseases and in that way prevent unnecessary loss of life and saved costs. Finally, the elimination of measles from Switzerland by 2020 requires such efforts on a broad scale [12].

## Figures and Tables

**Table 1 vaccines-07-00008-t001:** Participants’ socio-demographic characteristics.

Characteristics of Study Participants. N = 575	n	%
**Gender**		
Female	316	55
Male	259	45
**Country of birth**		
Switzerland	61	11
Europe	426	74
Africa	26	5
Asia	23	4
North America	22	4
South America	16	3
Oceania	1	0
**Contraction of the measles disease**		
Yes	209	36
No	251	44
I don’t know	115	20
**Year of birth**		
Before 1964	42	7
After 1964	533	93
**Possession of a vaccination record**		
Yes	369	65
No	202	35
**Vaccination against measles**		
Yes, had ≥ 2 doses	140	24
Yes, had 1 dose	80	14
No	95	17
Unknown	260	45

**Table 2 vaccines-07-00008-t002:** Predictive factors of measles immunization: multivariate analysis by logistic regression.

Variables	Adjusted Odds Ratio*	*P*	Confidence Interval 95%	
**Gender**				
Female	1	-	-	-
Male	**0.62**	**0.0051**	**0.43**	**0.86**
**Country of birth**				
Europe	1	-	-	-
Switzerland	**1.94**	**0.014**	**1.13**	**3.33**
Africa	**2.26**	**0.0393**	**1.01**	**5.05**
Asia	1.72	0.26	0.65	4.58
North America	2.33	0.04	0.98	5.55
South America	1.49	0.34	0.63	3.5
**Year of birth**				
Before 1964	1	-	-	-
After 1964	**3.28**	**0.0031**	**1.42**	**7.58**
**Contraction of measles disease**				
Yes	**1**	-	-	-
No	**6.01**	**<0.001**	**3.65**	**9.85**
I don’t know	1.53	0.12	0.87	2.67

* Adjusted on this variables: sex, country of birth, year of birth, Contraction of the measles disease.

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
