# Peer review of "Measles at Work: Status of Measles Vaccination at a Multinational Company"

_vaccines, 2019, doi:10.3390/vaccines7010008_

Round 1
Reviewer 1 Report
In 2012, the World Health Assembly endorsed the Global Vaccine Action Plan (GVAP), with the objective of eliminating measles by 2020. The estimated MCV1 coverage increased globally from 72% to 85%; annual reported measles incidence decreased 83%, from 145 to 25 cases per million population; and annual estimated measles deaths decreased 80%, from 545,174 to 109,638. Measles vaccination prevented an estimated 21.1 million deaths. However, measles elimination milestones have not been met, and three regions are experiencing a large measles resurgence. It is important that countries continue to strengthen case-based surveillance and increase MCV1 and MCV2 coverage and that immunization partners continue to raise the visibility of measles elimination goals and secure political commitment to these goals and sustained investments in health systems.
This work is an attempt to visualize certain pitfalls in vaccine uptake in a very constrainted environment. It might be locally relevant for the company tested but it lacks soundness. Some assumptions made are not cerainly reliable and backed by the results.
Improvement of references is needed.
Attacjhed the authors will find the manuscript with anotations to take into account and that are intended to improve quality.

Author Response
January 7, 2019.
Editorial Office of Vaccines
Dear Editor:
Thank you very much for your valuable comments on our article originally entitled “Measles at Work: Status of measles vaccination at a multinational company” submitted as an original contribution to Vaccines.
We value this opportunity to improve our manuscript and to answer to the reviewers’ comments. We use track change function in Microsoft Word for the revisions
We thank you for your consideration and look forward for our collaboration.
Referee: 1
Comment 1 : This paragraph is far too telegraphic. Please give more details as to we¡ther there are seroprevalence studies on measles antibody seroprevalence to back this affirmation on swiss citi¡zens not beong sufficiently immunized agains measles. Depending on age wild measles can be the immunizing agent , yet in a scenario of low measles virus circulation , vaccination is going to be responsible for positive immune status.
Answer 1 : Thank you for this comment, we have modified this paragraph with available information and data.
Comment 2 : According to Richard JL, Masserey Spicher V. Large measles epidemic in Switzerland from 2006 to 2009: consequences for the elimination of measles in Europe. Eur Surveill. 2009;14(50):pii: 19443. [PubMed]The outbreak occurred between 2006 and 2008. Please correct and include the reference. Another outbreak took place om Geneva in 2011 : This outbreak was the largest ever documented in the canton of Geneva. About one fifth of the cases were imported or related to imported cases. Reference :https://www.eurosurveillance.org/content/10.2807/ese.18.06.20395-en
Answer 2 : Thank you for this valuable comments. We have modified this paragraph and added these reference on the text.
Comment 3: Was this outnreak due to low vaccine uptake because of philosophical, religious reasons? What is the measles immunization schedule in Switzerland? Were there any special measures to enhance vaccination taken as a result of this outbreak. The outbreak lasted 10 years ? could you onform of the yearly distribution of cases. Were they all related?
Answer 3: We have modified this paragraph, because it did not adequately explain the facts we wanted to report.
Comment 4: A figure graph or table with the anual distribution of measles cases from 2006 should be included
Answer 4: Thank you, but we think it is not useful to add this graph for the comprehension of the article.
Comment 5: Is this subheading really needed?
Answer 5: yes we think that this subheading is needed for a better the comprehension of the article
Comment 6: Travelling to countries with low
endemicity and /or under elimination of measles status might not be
cadidate to travel clinic consultation. the phrase is ackward and
difficult to interpret. Do you mean that all travellers should consult
before departing, being whatsoever their destination?
Answer 6: We have modified this paragraph, because it did not adequately explain the facts we wanted to report.
Comment 7: This entire paragraph should be placed under the ethical approval section.
Answer 7: We have placed this entire paragraph under the ethical approval section
Comment 8: text is missing here ?
Answer 8: We do not think there is a lack of text, we wanted to know with this question whether the employees enrolled in the study were born before or after 1964, because those who were born before 1964 were most likely infected with measles in their childhood.
Comment 9: How was this clinical information assessed? Were there any consultations on registry or medical records ? There are many other viral diseases that can present with similar rash as measles , such as Dengue if they travelled too southeast asia or other dengue endemic countries.
Answer 9: This information was collected by self-declaration, there was no clinical Assessed for this information. This is a limitation in our data collection methodology.
Comment 10: How was the number of questionnaires to be sent out calculated ?
Answer 10 : We have not calculated how many questionnaires we must to send. All employees of this company received the questionnaire individually by mail.
Comment 11: During these 4 months, were there any refreshment messages for non-replyers?
The prevention campaign was only informative , so it was more an informative campaign . No preventive measure was undertaken . Correct?
Answer 11 : That’s exact, During these 4 months, there wat not refreshment messages for non-replyers,
This was not possible because Physician Company was not available for that during this period.
No preventive measure was undertaken. Correct?, yes it’s correct.
Comment 12: Did doctors administer the vaccine too? If not it's just a informative action again.
Answer 12: No, during this consultations with employees preparing their professional travel, doctor have administer measles vaccine for employees who requested it and/or were insufficiently vaccinated.
Comment 13: Include % of respondents. How many questionnaires were sent out , and how many respondents replied before recall if there was any recall for missing replies
Including a Data flow chart on measles survey would be more illustrating
Answer 13: we have included the % of respondents, there was not a recall mail.
Including a Data flow chart on measles survey would be more illustrating; We think that a flow chart is not useful, because there was only one email sent and no recall mail.
Comment 14: When did they contract the disease, while travelling or in childhood?
Answer 14: We have not this information, we don’t know if they contract disease during travelling or in childhood.
Comment 15: is there an immunization registry in Switzerland to be assessed and be able to verify their status'
Answer 15: no, unfortunately, there is no Immunization Registry in Switzerland that would allow us to know the status of people who are vaccinated, despite the fact that measles is a mandatory reportable disease in Switzerland.
Comment 16: This is a broad assumption, that should be statified by country. There are vast differences in vaccine coverage within Europe and even within the EU countries.
It might be more suitable to just have 2 categories , swiss and foreigners
Answer 16: That’s true, we could have done the analysis in stratified in 2 categories, Swiss and foreigners, other possibility was with 3 categories: swiss, Europe and other country, we chose to make this choice of categories stratified, because it was asked of us by the cantonal Health Service of Geneva.
Comment 17: This is not a very plaussible assumption, being that many countries u¡in the Africabn continent still have a high endemicity of measles. Besides the number is very low to infer statistical significance
Answer 17: That’s true, We’re not saying anything, this is just an observation related to our statistical analysis of the data. You’re probably right about the results we’ve found for African-native employees, so in the discussion we’ve been very cautious about how we interpret those results.
Comment 18: Was this information available for the study. A follow up could be carried out to assess the certainty of the data included in the quetsionnaire reply.
Answer 18: we have not this information about how many employees contacted the medical staff on site to clarify their personal situation after prevention campaign. It is true that it would have been very interesting to know how many employees took a blood test to know their immunity and how many go to their private doctor after the campaign, but unfortunately we do not know that information.
Comment 19: How many did get vaccinated as a result of the campaign?
Answer 19: We know that a number of employees were vaccinated by the company doctor following the information campaign, but unfortunately the company doctor cannot give us the exact number of employees vaccinated.
Comment 20: Is this reference correct? I see no information on costs stated .
Please include the source for this cost analysis
Answer 20: yes it’s a mistake, wa have added the right reference
Comment 21: There are far too many limitations to the study as is now.
Answer 21: We have updated limitations paragraph
Comment 22: need the reference for costs
Answer 22: We have added a reference for the cost.

Reviewer 2 Report
I suggest the author to add a reference to the paragraph 1.1 on Measles Swiss context. In particular it is not available a source of information for the 45 cases of measles in 2018, to which period those data are referred to?
Line 55. The sentence "It was sent confidentially to each of the..." is not finished. Please correct.
Line 82-83. Please check the entire sentence. I think there are some missing data..."44% said they had ????", is it something missing? In Table 1, 44% of participants said they had not contracted measles. Please check!
Author Response
Response to Referee: 2
Comment 1: I suggest the author to add a reference to the paragraph 1.1 on Measles Swiss context. In particular it is not available a source of information for the 45 cases of measles in 2018, to which period those data are referred to?
Answer 1: Thank you for this comment, we have add a reference to the paragraph 1.1 on measles swiss context.
Comment 2: Line 55. The sentence "It was sent confidentially to each of the..." is not finished. Please correct.
Answer 2: We have corrected this sentence.
Comment 3: Line 82-83. Please check the entire sentence. I think there are some missing data..."44% said they had ????", is it something missing? In Table 1, 44% of participants said they had not contracted measles. Please check!
Answer 3: Thank you for this comments. It’s a mistake, we have corrected this sentence.

Round 2
Reviewer 1 Report
I stil think that if the primary outcome of this survey is the complete measles immunization , a quantification of how many achieved this outcome should be included . I find it difficult to believe that the company's medical services don't have a registry of how many MMR vaccines vere administered as a result of the information delivered
I recomend that this data be included in the paper, at least an estimation
*In the abstract AOR should be aOR
Author Response
January 11, 2019.
Editorial Office of Vaccines
Dear Editor:
Thank you very much for your valuable comments on our article originally entitled “Measles at Work: Status of measles vaccination at a multinational company” submitted as an original contribution to Vaccines.
We value this opportunity to improve our manuscript and to answer to the reviewers’ comments. We use track change function in Microsoft Word for the revisions
We thank you for your consideration and look forward for our collaboration.
Referee: 1
Comment 1 : I stil think that if the primary outcome of this survey is the complete measles immunization , a quantification of how many achieved this outcome should be included
Answer 1 : we have modified methods paragraph to response of this comment. The number of employee which have achieved the complete measles immunization is included on the first part of the result paragraph
Comment 2 : I find it difficult to believe that the company's medical services don't have a registry of how many MMR vaccines vere administered as a result of the information delivered
I recomend that this data be included in the paper, at least an estimation
Answer 2 : The company's medical services was completely transformed between the study and the writing of this article. We were able to contact this doctor, who gave us the number of MMR vaccines performed post-intervention.
We included this data in the article
Comment 3: In the abstract AOR should be aOR
Answer 3: we have changed AOR for aOR in the abstract.
This manuscript has not been published elsewhere and is not under consideration by another journal. The corresponding author declares to have had full access to all the data in the study and takes responsibility for the integrity and accuracy of the data analysis. All authors have approved the manuscript and agree with this submission. The authors have no conflicts of interest to declare.
Yours sincerely,
Dr. Emilien Jeannot
